# A Scoring Tool to Predict Pulmonary Complications in Severe Leptospirosis with Kidney Failure

**DOI:** 10.3390/tropicalmed7010007

**Published:** 2022-01-11

**Authors:** Rizza Antoinette Yap So, Romina A. Danguilan, Eric Chua, Mel-Hatra I. Arakama, Joann Kathleen B. Ginete-Garcia, Joselito R. Chavez

**Affiliations:** 1Department of Adult Nephrology, National Kidney and Transplant Institute, Quezon City 1101, Philippines; radanguilan@gmail.com (R.A.D.); ericchuamd@gmail.com (E.C.); melhatraarakama@gmail.com (M.-H.I.A.); 2Department of Internal Medicine, National Kidney and Transplant Institute, Quezon City 1101, Philippines; joannkathleen@yahoo.com (J.K.B.G.-G.); chavezjrmd@yahoo.com (J.R.C.)

**Keywords:** mortality, predictors, pulmonary complications, pulmonary hemorrhage, severe leptospirosis

## Abstract

Rapid identification of patients likely to develop pulmonary complications in severe leptospirosis is crucial to prompt aggressive management and improve survival. The following article is a cohort study of leptospirosis patients admitted at the National Kidney and Transplant Institute (NKTI). Logistic regression was used to predict pulmonary complications and obtain a scoring tool. The Kaplan–Meir method was used to describe survival rates. Among 380 patients with severe leptospirosis and kidney failure, the overall mortality was 14%, with pulmonary hemorrhage as the most common cause. In total, there were 85 (22.4%) individuals who developed pulmonary complications, the majority (95.3%) were observed within three days of admission. Among the patients with pulmonary complications, 56.5% died. Patients placed on mechanical ventilation had an 82.1% mortality rate. Multivariate analyses showed that dyspnea (OR = 28.76, *p* < 0.0001), hemoptysis (OR = 20.73, *p* < 0.0001), diabetes (OR = 10.21, *p* < 0.0001), renal replacement therapy (RRT) requirement (OR = 6.25, *p* < 0.0001), thrombocytopenia (OR = 3.54, *p* < 0.0029), and oliguria/anuria (OR = 3.15, *p* < 0.0108) were significantly associated with pulmonary complications. A scoring index was developed termed THe-RADS score (Thrombocytopenia, Hemoptysis, RRT, Anuria, Diabetes, Shortness of breath). The odds of developing pulmonary complications were 13.90 times higher among patients with a score >2 (63% sensitivity, 88% specificity). Pulmonary complications in severe leptospirosis with kidney failure have high mortality and warrant timely and aggressive management.

## 1. Introduction

Leptospirosis is a worldwide public health problem commonly encountered in humid tropical and subtropical areas where most developing countries are found [1]. It is caused by the pathogenic species of Leptospira, commonly *L. interrogans*, with *L. icterohemorrhagica* causing the severe form [2]. This condition results upon exposure of the mucous membranes to the urine of infected animals, usually rats, and can manifest with a wide range of symptoms, from a mild flu-like illness to more severe complications such as jaundice, meningitis, renal dysfunction, and hemorrhage [3].

In the Philippines, an upsurge of leptospirosis cases is commonly seen during flooding and heavy rainfall by bringing bacteria and their animal hosts into closer contact with humans [4]. According to the government’s Department of Health, 1227 leptospirosis cases were recorded in Metro Manila from January to August 2018. This is a 358% increase from the 268 recorded cases during the same period in 2017 [5]. The devastation brought about by typhoon Ketsana in 2009 prompted NKTI to set up a ward dedicated to leptospirosis patients to accommodate more than 60 patients at a single period [6]. Overall mortality was 15.6% due mainly to pulmonary hemorrhage [7].

In recent years, the incidence of a severe pulmonary form of leptospirosis (SPFL) ranges from 20% to 70%. It has a case fatality rate of >50%, significantly higher than that for Weil’s disease (a triad of jaundice, renal failure, and bleeding), which is 5–15% [8]. Pulmonary involvement in leptospirosis has increased in the last few years [9,10], with coughing, hemoptysis, and dyspnea as the most common symptoms [11]. Pulmonary symptoms usually appear between the fourth and sixth day of illness and may rapidly evolve to death in less than 72 h [12].

Pulmonary hemorrhage and acute respiratory distress syndrome (ARDS), common pulmonary complications of leptospirosis, are associated with high mortality [13]. Profuse lung hemorrhaging may be due to capillary involvement and thrombocytopenia [12]. Its pathogenesis has yet to be clarified, and several causal factors are suggested, such as direct leptospiral invasion resulting in capillaritis, the immune host response, and environmental conditions [14].

Due to its high case fatality rate, there is a need to quickly identify patients at risk of developing pulmonary complications since they require intensive monitoring, early pulmonary invasive intervention, and aggressive supportive care [15]. This study will identify predictors of pulmonary complications in a high-risk population so that intensive monitoring and timely intervention can lead to more lives saved.

### Objectives

The general objective of the study was to identify predictors of pulmonary complications in patients with severe leptospirosis. Specifically, the study aimed to: Determine the proportion of patients with severe leptospirosis who developed pulmonary complications.Determine the proportion of patients with early (≤3 days of admission) versus late (>3 days after admission) presentation of pulmonary complications.Describe and compare the baseline demographic and clinical characteristics of patients with severe leptospirosis among those with and without pulmonary complications.Describe the respiratory support of patients according to ARDS severity (mild: <100, moderate: 100–200, severe: >200).Compare the proportion of the patients who developed the following conditions during their hospital stay among those with and without pulmonary complications: Acute kidney injury (AKI), pancreatitis, bleeding, cardiac complications, neurological complications.Compare the number of intensive care unit (ICU) days, dialysis days, length of hospital stay, use of inotropes, number of blood products transfused, and renal replacement therapy (RRT) requirements among those with and without pulmonary complications.Determine overall survival among those who developed pulmonary complications until hospital discharge.Among those with ARDS, compare survival on extracorporeal membrane oxygenation (ECMO) versus mechanical ventilation alone.Develop a weighted scoring index and determine hazard ratios for each parameter.

## 2. Materials and Methods

### 2.1. Study Design

A retrospective cohort was used to determine the predictors of pulmonary complications in severe leptospirosis.

### 2.2. Study Area and Population

Records/charts of patients diagnosed with leptospirosis at NKTI from 1 January to 31 December 2018 were reviewed.

### 2.3. Inclusion Criteria

Patients above 18 years of age.Patients diagnosed with leptospirosis and admitted to NKTI based on clinical manifestations according to the 2010 Philippine Clinical Practice Guidelines on the Diagnosis, Management, and Prevention of Leptospirosis [13].Patients with pulmonary complications.

### 2.4. Exclusion Criteria

Patients with a previous history of chronic kidney disease.Patients with a previous history of AKI requiring RRT.Patients with a previous history of congestive heart failure, stroke, myocardial infarction, and chronic obstructive pulmonary disease that require oxygen.Patients who have blood dyscrasias or hematologic problems.Patients with pre-existing lung disease, requiring continuous oxygen therapy.

Data such as demographic characteristics, clinical signs and symptoms, and laboratory findings were obtained from patient charts. Those presenting with pulmonary complications were also determined. Mortalities were recorded.

During hospitalization, all patients received antimicrobial therapy with penicillin or ceftriaxone. When indicated, patients received intravenous fluid resuscitation, vasoactive drugs, appropriate respiratory support, blood component transfusions, and ICU monitoring. Renal replacement therapy was initiated for indications of oliguria, hyperkalemia, severe acidosis, fluid overload, and uremic syndrome. Based on the fulfillment of our institution’s protocol for pulse therapy, IV methylprednisolone was given [2,6,16]. Cyclophosphamide was also given for episodes of hemoptysis.

Laboratory tests for leptospirosis were performed. Serum samples were collected at the time of hospital admission. A serum antibody test with Leptospira IgM ELISA assay was conducted. Samples were also sent to the Research Institute of Tropical Medicine (RITM) for a microscopic agglutination test (MAT) and detection of Leptospira species DNA via PCR tests.

### 2.5. Definition of Terms

Suspected Leptospirosis Case—patients with at least 2 days of acute febrile illness AND either reside in a flooded area with a high-risk exposure (wading in floodwaters, bitten by rats, work in sewerage areas) or presenting with at least two of the following symptoms: myalgia, calf tenderness, conjunctival suffusion, chills, abdominal pain, headache, jaundice, or oliguria [13].

Severe Leptospirosis—the presence of unstable vital signs, jaundice/icteric sclera, abdominal pain, nausea, vomiting and diarrhea, oliguria/anuria, shock, meningismus, altered mental status, dyspnea, or hemoptysis [13].

Pulmonary Complications—the presence of both of the following [15]:

At least two of the following clinical signs: dyspnea, cough, chest pain, and crackles on auscultation.

Radiographic abnormalities: parenchymal, interstitial, and alveolar infiltrates or densities.

Respiratory support—defined according to the most invasive pulmonary support received prior to demise or recovery.

ARDS severity—defined according to the Berlin definition [17]. The patient classification was designated according to the most severe partial pressure of oxygen (PO_2_)/fraction of inspired oxygen (FIO_2_) (P/F) ratio obtained via arterial blood gas (ABG) analysis. Stable patients not given oxygen support and without any ABG results with oxygen saturation of ≥95% via peripheral pulse oximeters were assumed to have a (pO_2_) of at least 80 (normal pO_2_ 80–100 mmHg) and FIO_2_ of 21% (room air).

Acute Kidney Injury—defined according to the 2012 Kidney Disease Improving Global Outcomes Clinical Practice Guideline for Acute Kidney Injury [18].

Pancreatitis—diagnosed by elevated amylase and lipase levels three times the normal level along with clinical symptoms.

Bleeding—defined as any significant blood loss either internally or externally (at least one of the following: hemoptysis, gastrointestinal hemorrhage, purpura, conjunctival bleeding, hematuria, epistaxis) [19].

Cardiac Complications—defined as the presence of ischemic heart disease or heart failure.

Neurological Complications—defined as the presence of meningitis, seizures, cognitive impairment, encephalopathy, transient ischemic attacks, or stroke.

### 2.6. Sampling and Sample Size

Total enumeration or a review of all eligible patient charts from January 2018 to December 2018, was conducted due to the considerable number of predictors considered in the study. A data abstraction tool was utilized by investigators to collect relevant data from the included patient charts.

### 2.7. Data Analysis

A coding manual was used to guide the encoding process. The generated data set was exported to STATA 14 for processing and analysis. Mean, or median was used to summarize continuous data. An unpaired t-test was used to test for the difference of means. Categorical data were presented as number and proportion (n [%]). A Chi-Square Test or Fisher Exact Test was used to test the difference in proportion. Kaplan–Meier Curves were used to graphically describe the survival rates. Univariate and multivariate logistic regression were used for a model to predict the occurrence of pulmonary complications. This model was also used to formulate the scoring tool.

Additionally, the hazard ratio was provided by utilizing univariate and multivariate cox proportional-hazards regression. The level of significance was 5%.

### 2.8. Ethical Considerations

Upon approval of the Technical Review Board, the protocol was submitted to the Research Ethics Committee of NKTI and was issued ethics clearance. Only the investigators and data collectors had access to the patient charts. A unique code number was assigned to each chart abstracted.

## 3. Results

Among 380 clinically diagnosed leptospirosis patients admitted in NKTI from January to December 2018, 142 were positive in the IgM Leptospiral antibody test, 64 were positive in the MAT, and 56 were positive in the real-time PCR detection of pathogenic Leptospira spp. Out of all these patients, 85 (22.4%) developed pulmonary complications.

The majority were male with a mean age of 38 years ± 13.3. A total of 95.3% developed pulmonary complications within three days of admission. There was no significant difference in age (*p* = 0.15), gender (*p* = 0.22), floodwater exposure (*p* = 0.90), and occupation (*p* = 0.23) between those who had pulmonary complications or not (Table 1).

Univariate analysis showed that the presence of diabetes (*p* ≤ 0.0001), jaundice (*p* = 0.0133), dyspnea (*p* ≤ 0.0001), and hemoptysis (*p* ≤ 0.0001) significantly predicted pulmonary complications. Among the baseline laboratory examinations thrombocytopenia (*p* = 0.0002), low potassium (*p* = 0.0094), high phosphorous (*p* = 0.0394), high creatinine (*p* = 0.0295), high BUN (*p* = 0.0439), congestion on chest X-ray (*p* = 0.0035), metabolic acidosis (*p* ≤ 0.0001), respiratory acidosis (*p* = 0.0203), hypoxemia (*p* = 0.0062), and oliguria/anuria (*p* = 0.0001) were also predictive of pulmonary complications (Table 1).

The severity of the respiratory failure and the oxygen support required by leptospirosis patients were analyzed (Table 2). The majority (62.1%) did not require any respiratory support and had a P/F ratio of more than 200. Twenty-five patients were given oxygen support via nasal cannula (6.6%), while twenty-six patients required oxygen via face mask (6.8%). Thirty-eight patients (10%) required invasive mechanical ventilation, and five intubated patients were placed on ECMO (1.3%).

Univariate analysis showed that pulmonary complications were highest among patients who also presented with the following complications: pancreatitis (*p* = 0.0224), bleeding (*p*-value < 0.001), and cardiac complications (*p* = 0.0001) (Table 3). Mean number of ICU days (*p* = 0.0004), number of dialysis days (*p* = 0.040), length of hospital stay (*p* = 0.002), mean number of blood products transfused (*p* ≤ 0.0001), and the need for hemodialysis (*p* ≤ 0.0001) were significantly higher among those with pulmonary complications (Table 4).

The overall mortality in this population was 14%, and it increased significantly (*p* < 0.0001) among patients with pulmonary complications. The odds of dying were 94.38 times higher in patients with pulmonary complications (95% CI 32.19–276.76). This lower survival probability was evident in the Kaplan–Meier curve, where their median survival was only 6 days (95% 2.0 to 32.0, *p* < 0.0001) (Figure 1). The P/F ratios of the leptospirosis patients were cross tabulated against respiratory support and survival (Table 5). Among those with a P/F ratio < 100, 23 (82.1%) on mechanical ventilation died compared to a mortality of only 3.6% among those placed on ECMO (*p* = 0.0004), although only five patients were put on ECMO.

Univariate analysis indicated that diabetes, dyspnea, hemoptysis, thrombocytopenia, abnormal chest X-ray, oliguria/anuria, jaundice, and RRT requirement were associated with pulmonary complications. After multivariate logistic regression, only RRT requirement, diabetes, dyspnea, hemoptysis, thrombocytopenia and oliguria/anuria remained significant predictors (Table 6). This final model was the basis for the scoring index to predict pulmonary complications in severe leptospirosis patients termed THe-RADS score (Thrombocytopenia, Hemoptysis, RRT, Anuria, Diabetes, Shortness of breath). Specifically, those with diabetes, dyspnea, and hemoptysis were given a score of 2 each, while RRT requirement, thrombocytopenia, and oliguria/anuria were each given a score of 1. The resulting index had a maximum score of 9. If a score of >2 is obtained, the patient has a 62% chance of developing pulmonary complications (Figure 2).

The area under the receiver operating characteristic curve of this scoring index had high diagnostic accuracy and was statistically significant with an area under the curve (AUC) of 0.841 (*p* < 0.001) (Figure 3). A cut-off score of >2 resulted in 63.41% (95% CI 52.0% to 73.8%) sensitivity and 88.57% (95% CI 84.3 to 92.1%) specificity (Table 7). This indicates that a score of >2 portends a 13.90 times higher risk of developing pulmonary complications.

Multivariate cox regression analysis showed that dyspnea, diabetes, RRT requirement, hemoptysis, thrombocytopenia, and oliguria/anuria were predictive of pulmonary complications based on their hazard ratios (Table 8). If a leptospirosis patient has any one of these factors, the risk of developing pulmonary complications was 5.13, 3.21, 3.21, 2.72, 2.23, and 1.63, respectively. The more factors a patient had, the higher the summed-up hazard ratio.

## 4. Discussion

This study reviewed 380 severe leptospirosis patients admitted to NKTI in 2018, where 85 (22.4%) patients developed pulmonary complications. This was higher than the reported incidence in a Thailand study where only 5.8% had pulmonary complications [11]. The incidence of pulmonary involvement in leptospirosis varies from 20 to 70% [8]. The majority of patients in this study (95.3%) developed complications within three days of admission. In a Sri Lankan study in 2017, patients developed pulmonary symptoms within the first week of illness, mostly on days 4 and 5 [20]; thus, pulmonary complications develop early during hospital admission.

Most patients were male with a mean age of 38 years with a flood exposure history. There is an increased chance for males to be exposed to the etiologic agent for leptospirosis due to occupational risk [21]. Most patients with pulmonary complications were between 18 and 40 years (54%) because 60% of the population in this study belong to this age bracket. Notably, in this study, 40% of patients did not have exposure to floods, and 58% did not have high-risk occupations but still developed complications and death. In a 2009 meta-analyses, certain studies stated that there was no significant association of leptospirosis with flooding and occupation [21].

The involvement of the lungs varies from subtle clinical features to deadly pulmonary hemorrhage and ARDS [22]. In this study, as the patient’s oxygen requirement increased, the P/F ratio correspondingly decreased, and respiratory support became more invasive. However, there were some exceptions. Three patients were hooked to mechanical ventilation with a P/F ratio of > 200 due to decreased sensorium, secondary to uremia and severe respiratory distress from metabolic acidosis and congestion. Three patients with a P/F ratio of <100 were hooked to *bilevel positive airway pressure* (BiPAP) due to normal sensorium and were immediately dialyzed. Hence, decisions for respiratory support should not only depend on the P/F ratio but also on clinical evaluation. 

In a review by Gulati et al., most leptospirosis cases showed alveolar hyperventilation with hypocapnia. Hypoxemia was observed in 75% of the patients, probably due to pulmonary veno-arterial shunts in impaired pulmonary areas. Patients with a combination of oliguric renal failure, pulmonary abnormalities, and chest radiograph involvement were also shown to have lower PaO2 values [22]. Thus, the presence of severe pulmonary involvement in leptospirosis is complicated by disease involving other organs, especially the kidneys. 

In a study by Paganin et al., 44 (33%) of the leptospirosis patients with pulmonary involvement had oliguria/anuria, and this was found to be an independent factor related to severe pulmonary involvement. The combination of AKI with respiratory impairment presents a risk of lethal evolution: mortality is 18% for AKI and 24% for respiratory impairment alone, but 55% when the two are combined [15]. Thus, patients entering with severe leptospirosis should be investigated for the presence of both renal and lung complications. Moreover, patients presenting with either one as their primary complaint should be investigated for the other complications as well. In our institution, all patients with leptospirosis are provided with a chest X-ray and ABG as part of standard care.

In a study conducted in the Reunion Islands, the use of mechanical ventilation was strongly correlated with mortality [15]. This was also seen in our study, where patients with severe ARDS on mechanical ventilators had the highest mortality at 82.1%. Five intubated patients were hooked to ECMO, which resulted in a mortality of only 3.6%. Although numbers were small, ECMO is a promising modality in increasing patient survival. In 2019, Vandroux et al. reported eight leptospirosis patients who underwent ECMO for refractory ARDS, and 75% survived [23].

Mortality rates of pulmonary involvement in leptospirosis vary in different studies. In a study by Marotto et al. in 2010, 25% of 203 leptospirosis patients developed leptospirosis-associated pulmonary hemorrhage syndrome (LPHS) with an 8% mortality rate [24]. More deaths were observed in the 2011 study by Paganin, where overall mortality was 15.7% (21 patients). In total, 40 patients had severe pulmonary leptospirosis with ARF, and 16 of them died (40%). The study suggested that the rapid urbanization and expanding urban poverty, together with climate, was associated with severe pulmonary forms of leptospirosis and led to death [15]. 

In a recent study by Vandroux in 2019, among 172 leptospirosis patients admitted from 2004 to 2015, 39 patients (23%) presented with ARDS with a mortality rate of 23% (9 cases). This study concluded that the prognosis of leptospirosis-related ARDS was better compared to other causes of ARDS [23]. 

In the Philippines, groundbreaking numbers of leptospirosis were recorded yearly due to outbreaks during the monsoon season. In a 2014 study by Manipol et al. also in NKTI, among 138 patients admitted from 2009–2013, 26 (18%) patients died, primarily due to pulmonary hemorrhage [16]. In a later review by Pasamba et al. in 2018 in the same institution, among 194 leptospirosis patients from 2004–2016, 49 patients (25%) died, 16% due to pulmonary hemorrhage [7]. All of these studies suggest that pulmonary complications are responsible for the high mortality seen in severe leptospirosis.

In 2018, with another leptospirosis outbreak, this single-center study reported 380 leptospirosis patients with an overall mortality rate of 14% (52 patients). Out of the 85 (22.4%) who had pulmonary complications, 48 (56.5 %) died, and the median survival time was 6 days (95% 2.0 to 32.0, *p* <0.0001). The most common cause of death was respiratory failure due to diffuse pulmonary hemorrhage in a population where the majority required renal replacement. 

Another reason for the higher mortality rate in this study is the fact that NKTI, a tertiary referral center, has a higher number of severe leptospirosis cases presenting with both pulmonary complications and dialysis-requiring AKI. In a Brazilian study, increased serum creatinine in patients with LPHS was identified as an independent risk factor, but there was no mention if their patients presented with oliguria, anuria, or underwent dialysis [22]. In a Reunion Islands study, oliguria/anuria was found to be an independent factor related to severe pulmonary involvement, which further increased mortality [15].

The major finding of this study was that dyspnea, hemoptysis, diabetes, RRT requirement, thrombocytopenia, and oliguria/anuria were significant independent predictors of pulmonary complications in patients with severe leptospirosis. It is important to identify factors predictive of rapid deterioration in patients that warrant intensive monitoring and aggressive management.

To our knowledge, this is the first study providing a scoring index to identify patients with severe leptospirosis at risk for developing pulmonary complications. Although dyspnea and oliguria/anuria were identified in the study by Paganin et al. to be independent predictors of pulmonary involvement in leptospirosis, there was no scoring system to quantify the relative risk of severe disease [15].

Marotto et al. developed a multivariate predictive model for LPHS and identified five factors: serum potassium, serum creatinine, respiratory rate, shock, and Glasgow Coma Scale Score < 15. The scoring system was based on multiplying the natural logarithm of the odds ratio for each risk factor identified in the logistic regression analysis and rounding the value to the nearest integer [24]. This may prove to be tedious in the emergency room setting, wherein rapid decisions need to be made. Therefore, using the simple scoring system developed in this study is advantageous, and a cut-off of >2 will identify leptospirosis patients at risk for pulmonary complications with 63.41% sensitivity and 88.57% specificity.

Limitations of the study include its retrospective nature. The scoring index will also require external validation. Since the majority of the patients were on dialysis, the scoring system may be applicable only to this high-risk population.

## 5. Conclusions

Among the 380 patients diagnosed with severe leptospirosis, 22.4% developed pulmonary complications within 3 days. Most patients with pulmonary complications had a P/F ratio of >200, and only 30% required respiratory support.

Pulmonary complications increased in patients with concomitant pancreatitis (*p* = 0.0224), bleeding (*p*-value < 0.001), and cardiac complications (*p* = 0.001). The mean number of ICU days (*p* = 0.0004), number of dialysis days (*p* = 0.040), length of hospital stay (*p* = 0.002), mean number of blood products transfused (*p* ≤ 0.0001), and need for hemodialysis (*p* ≤ 0.0001) were also significantly higher among those with pulmonary complications

Patients with pulmonary complications had a mortality of 56% with a median survival of 6 days (95% 2.0 to 32.0, *p* < 0.0001). Those on mechanical ventilation had an 82.1% mortality, and this dropped to only 3.6% for the patients on ECMO.

THe-RADS score was derived from multivariate analyses showing dyspnea (OR = 28.76, *p* < 0.0001), hemoptysis (OR = 20.73, *p* < 0.0001), diabetes (OR = 10.21, *p* < 0.0001), RRT requirement (OR = 6.25, *p* < 0.0001), thrombocytopenia (OR = 3.54, *p* < 0.0029), and oliguria/anuria (OR = 3.15, *p* < 0.0108) to be significantly associated with pulmonary complications. A score of >2 accurately predicted death with a 63.41% sensitivity and 88.57% specificity. The same six parameters increased the risk of developing pulmonary complications 2–5 times based on their hazard ratios.

Leptospirosis remains a devastating illness. Early diagnosis and timely intervention of pulmonary complications may improve survival. This simple six-point clinical score can provide the clinician with a reliable tool to predict early pulmonary complications that will warrant aggressive management that may save more lives.

## Figures and Tables

**Figure 1 tropicalmed-07-00007-f001:**
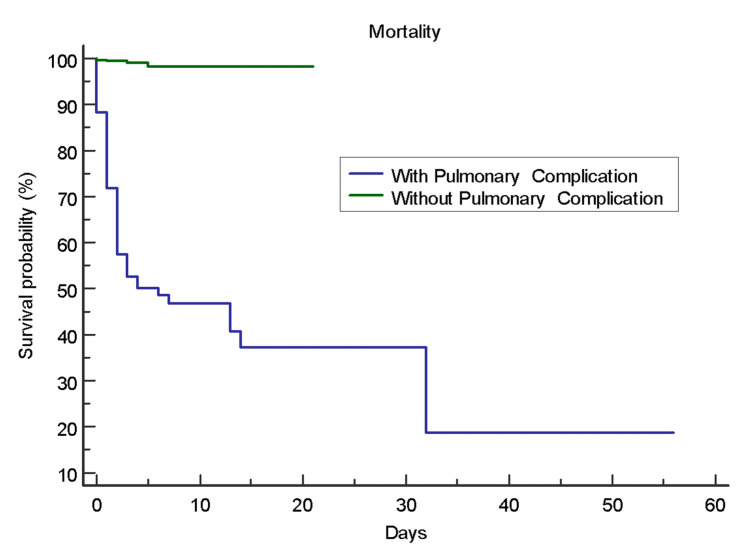
Kaplan–Meier survival curve.

**Figure 2 tropicalmed-07-00007-f002:**
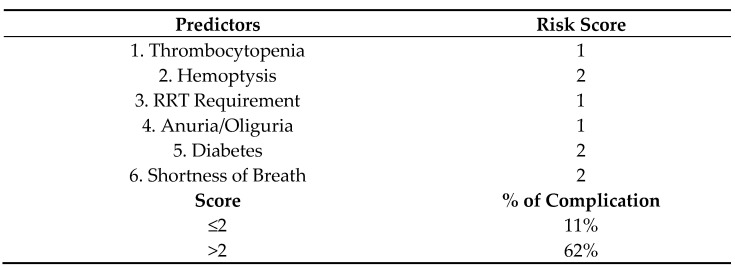
Scoring tool: THe-RADS score.

**Figure 3 tropicalmed-07-00007-f003:**
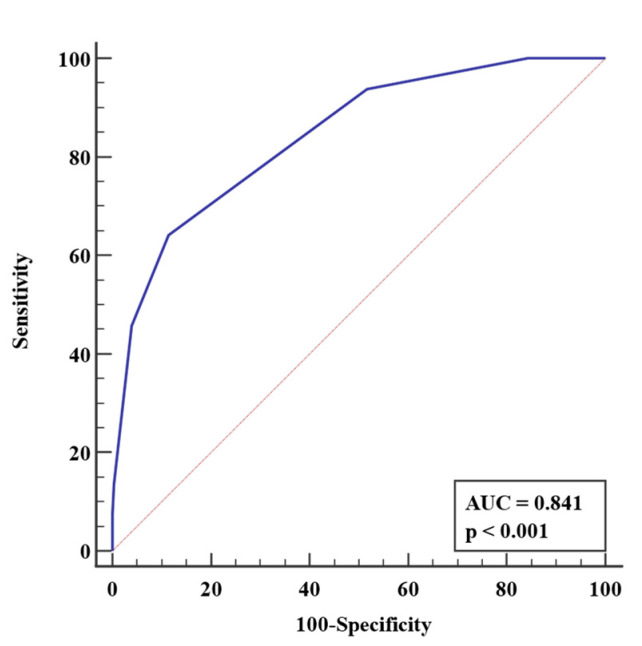
Receiver operating curve of scoring index in predicting pulmonary complications.

**Table 1 tropicalmed-07-00007-t001:** Distribution of pulmonary complications according to baseline characteristics.

Characteristics	Pulmonary Complication; n (%)		Total	*p*-Value
	without (n = 295)	with (n = 85)		
Age (n = 380)				
18–40	183 (62.0)	46 (54.1)	229	0.1466
41–60	99 (33.6)	31 (36.5)	130	
61 and above	13 (4.4)	8 (9.4)	21	
Sex (n = 380)				
Female	47 (15.9)	9 (10.6)	56	0.2213
Male	248 (84.1)	76 (89.4)	324	
Flood water exposure history (n = 380)				
Without	116 (39.7)	34 (40.5)	150	0.9017
With	176 (60.3)	50 (59.5)	226	
*Occupation (n = 380)				
Low risk	191 (65.4)	49 (58.3)	240	0.2348
High Risk	101 (34.6)	35 (41.7)	136	
Diabetes				
Absent	273 (92.5)	51 (60.0)	324	≤0.0001
Present	22 (7.5)	34 (40.0)	56	
Hypertension				
Absent	280 (94.9)	78 (91.8)	358	0.2738
Present	15 (5.1)	7 (8.2)	22	
Clinical Presentation				
Fever (n = 380)				
Absent	19 (6.4)	4 (4.7)	23	0.5551
Present	276 (93.6)	81 (95.3)	357	
Jaundice (n = 380)				
Absent	268 (90.8)	69 (81.2)	337	0.0133
Present	27 (9.2)	16 (18.8)	43	
Dyspnea (n = 380)				
Absent	292 (99.0)	77 (90.6)	369	≤ 0.0001
Present	3 (1.0)	8 (9.4)	11	
Hemoptysis (n = 380)				
Absent	291 (98.6)	73 (85.9)	364	≤ 0.0001
Present	4 (1.4)	12 (14.1)	16	
Hypotension (n = 380)				
Absent	289 (98.0)	81 (95.3)	370	0.2402
Present	6 (2.0)	4 (4.7)	10	
Baseline laboratory values				
*WBC/Leukocytes (n = 365)				
Normal	103 (36.4)	21 (25.6)	124	0.0698
Abnormal	180 (63.6)	61 (74.4)	241	
Platelets (n = 363)				
Normal	106 (37.7)	13 (15.9)	119	0.0002
Abnormal	175 (62.3)	69 (84.1)	244	
Prothrombin TIme (n = 348)				
Normal	82 (30.3)	25 (32.5)	107	0.7112
Abnormal	189 (69.7)	52 (67.5)	241	
Partial Thromboplastin Time (n = 349)				
Normal	37 (13.6)	12 (15.6)	49	0.6591
Abnormal	235 (86.4)	65 (84.4)	300	
Potassium (n = 374)				
Normal	156 (53.4)	57 (69.5)	213	0.0094
Abnormal	136 (46.6)	25 (30.5)	161	
Calcium (n = 256)				
Normal	24 (12.4)	2 (3.2)	26	0.0506
Abnormal	170 (87.6)	60 (96.8)	230	
Magnesium (n = 187)				
Normal	70 (47.9)	18 (43.9)	88	0.6476
Abnormal	76 (52.1)	23 (56.1)	99	
Phosphorous (n = 57)				
Normal	30 (60.0)	1 (14.3)	31	0.0394
Abnormal	20 (40.0)	6 (85.7)	26	
*SGPT (n = 70)				
Normal	19 (32.8)	8 (61.5)	27	0.0653
Abnormal	39 (67.2)	5 (38.5)	44	
Creatinine (n = 365)				
Normal	17 (5.9)	0 (0.0)	17	0.0295
Abnormal	269 (94.1)	79 (100.0)	348	
Blood Urea Nitrogen (n = 261)				
Normal	13 (6.4)	0 (0.0)	13	0.0439
Abnormal	189 (93.6)	60 (100.0)	249	
Chest X-ray (n = 361)				
Without pulmonary congestion	129 (46.6)	24 (28.6)	153	0.0035
With pulmonary congestion	148 (53.4)	60 (71.4)	208	
*KUB (n = 91)				
Without parenchymal disease	9 (12.2)	3 (17.6)	12	0.6903
With parenchymal disease	65 (87.8)	14 (82.4)	79	
Metabolic Acidosis (n = 285)				
Without	133 (64.9)	29 (36.2)	162	≤ 0.0001
With	72 (35.1)	51 (63.7)	123	
Respiratory Acidosis (n = 285)				
Without	204 (99.5)	76 (95.0)	280	0.0203
With	1 (0.5)	4 (5.0)	5	
Hypoxemia (n = 279)				
Without	179 (89.9)	62 (77.5)	241	0.0062
With	20 (10.1)	18 (22.5)	38	
Urine Output (n = 375)				
Normal	171 (58.6)	38 (45.8)	209	0.0001
Oliguric	104 (35.6)	27 (32.5)	131	
Anuric	17 (5.8)	18 (21.7)	35	

*High-risk occupations: drivers, market/sidewalk vendors, garbage collectors, security guards, messengers, farmers, carpenters, plumbers, construction workers, fishermen; *WBC: white blood cell; *SGPT: serum glutamic pyruvic transaminase test; *KUB: kidney, ureter, urinary bladder ultrasound.

**Table 2 tropicalmed-07-00007-t002:** Respiratory support required among leptospirosis patients according to P/F Ratio.

Table.	*P/F Ratio (n)			Total (n = 380)
	<100 (n = 41)	100–200 (n = 70)	>200 (n = 269)	
None	0	33	236	269
Oxygen support via nasal cannula	0	7	18	25
Facemask	1	15	10	26
*BiPAP	9	6	2	17
Mechanical ventilator only	26	9	3	38
*ECMO	5	0	0	5

*P/F Ratio: ratio of arterial oxygen partial pressure to fractional inspired oxygen; *BiPAP: bilevel positive airway pressure; *ECMO: extracorporeal membrane oxygenation.

**Table 3 tropicalmed-07-00007-t003:** Distribution of pulmonary outcome according to complications.

Complications	Pulmonary Complication; n (%)		Total	*p*-Value
	without (n = 295)	W44ith (n = 85)		
Acute kidney injury (n = 380)				
Absent	26 (8.8)	8 (9.4)	34	0.8650
Present	269 (91.2)	77 (90.6)	346	
Pancreatitis (n = 380)				
Absent	293 (99.3)	81 (95.3)	374	0.0224
Present	2 (0.7)	4 (4.7)	6	
Bleeding (n = 380)				
Absent	287 (97.3)	29 (34.1)	316	< 0.0001
Present	8 (2.7)	56 (65.9)	64	
Cardiac complications (n = 380)				
Absent	295 (100.0)	79 (92.9)	374	0.0001
Present	0 (0.0)	6 (7.1)	6	
Neurological complications (n = 380)				
Absent	294 (99.7)	83 (97.6)	377	0.1270
Present	1 (0.3)	2 (2.4)	3	

**Table 4 tropicalmed-07-00007-t004:** Distribution of pulmonary outcome according to treatment-related data.

Table.	without Pulmonary Complications			with Pulmonary Complications			*p* value
	n	Mean	SD	n	Mean	SD	
Number of *ICU Days	295	0	0	84	1.11	5.35	0.0004
Number of Dialysis Days	220	2.03	0.99	79	3.01	5.66	0.0140
Length of Hospital Stay	295	4.56	2.51	85	6.84	9.31	0.0002
Number of Inotropes	295	0.02	0.15	85	0.04	0.19	0.5585
Number of Blood Products Transfused	295	0.21	1.22	85	9.09	31.99	≤0.0001
	n	%		n	%		
Mode of *RRT							
None	8	9.5		77	26.4		≤0.0001
Hemodialysis	75	89.3		187	64.0		
Peritoneal Dialysis	1	1.2		27	9.6		

*ICU: intensive care unit, *RRT: renal replacement therapy.

**Table 5 tropicalmed-07-00007-t005:** Cross-tabulation of P/F ratio and mortality.

	Survived	Died	Total	*p*-Value
Among *P/F > 200				
Nasal Cannula	15 (7.6)	3 (33.3)	18 (8.7)	0.0001
Face Mask	8 (4.1)	1 (11.1)	9 (4.4)	
*BiPAP	1 (.5)	0 (0.0)	1 (.5)	
Mechanical Ventilator	0 (0.0)	3 (33.3)	3 (1.5)	
None	173 (87.8)	2 (22.2)	175 (85)	
Among P/F 100–200				
Nasal Cannula	6 (10.5)	1 (7.7)	7 (10)	0.0001
Face Mask	13 (22.8)	1 (7.7)	14 (20)	
BiPAP	6 (10.5)	1 (7.7)	7 (10)	
Mechanical Ventilator	0 (0.0)	9 (69.2)	9 (12.9)	
None	32 (56.1)	1 (7.7)	33 (47.1)	
Among P/F < 100				
Face Mask	0 (0.0)	1 (3.6)	1 (2.4)	0.0004
BiPAP	6 (46.2)	3 (10.7)	9 (22)	
Mechanical Ventilator	3 (23.1)	23 (82.1)	26 (63.4)	
*ECMO	4 (30.8)	1 (3.6)	5 (12.2)	

*P/F Ratio: ratio of arterial oxygen partial pressure to fractional inspired oxygen; *BiPAP: bilevel positive airway pressure; *ECMO: extracorporeal membrane oxygenation.

**Table 6 tropicalmed-07-00007-t006:** Multivariate regression analysis (final model) for the Scoring Index of Pulmonary Complication *.

Factors	Multivariate					Score
	Coefficient	Standard Error	Adjusted OR	95% CI	*p*-Value	
Mode of *RRT						
None/*Peritoneal dialysis	Reference					
Hemodialysis	1.8320	0.48037	6.25	2.44 to 16.01	0.0001	1
Diabetes						
Absent						
Present	2.3234	0.37984	10.21	4.85 to 21.50	0.0001	2
Dyspnea						
Absent	Reference					
Present	3.359	0.83286	28.76	5.62 to 147.14	0.0001	2
Hemoptysis						
Absent	Reference					
Present	3.0318	0.75728	20.73	4.70 to 91.47	0.0001	2
Platelet						
Normal	Reference					
Abnormal	1.2645	0.42517	3.54	1.54 to 8.15	0.0029	1
Urine Output						
Normal/*Oliguric	Reference					
Anuric	1.1482	0.45031	3.15	1.30 to 7.62	0.0108	1

*RRT: renal replacement therapy, * Jaundice and Chest X-ray were removed since it becomes nonsignificant during the multivariate run, *Peritoneal dialysis was merged with None (mode of RRT), as it is also non-significant, *Oliguric was merged with Normal (Urine), as it is also non-significant.

**Table 7 tropicalmed-07-00007-t007:** Diagnostic values at different cut-off score.

Criterion	Sensitivity	95% CI	Specificity	95% CI	+LR	95% CI	−LR	95% CI
≥0	100	95.6–100.0	0	0.0–1.3	1	1.0–1.0		
>0	100	95.6–100.0	15.71	11.7–20.5	1.19	1.1–1.2	0	
>1	93.9	86.3–98.0	48.21	42.2–54.2	1.81	1.6–2.1	0.13	0.05–0.3
>2	63.41	52.0–73.8	88.57	84.3–92.1	5.55	3.9–8.0	0.41	0.3–0.6
>3	45.12	34.1–56.5	96.07	93.1–98.0	11.49	6.1–21.5	0.57	0.5–0.7
>4	13.41	6.9–22.7	99.64	98.0–100.0	37.56	4.9–286.7	0.87	0.8–0.9
>5	7.32	2.7–15.2	100	98.7–100.0			0.93	0.9–1.0
>7	0	0.0–4.4	100	98.7–100.0			1	1.0–1.0
**Score**		**n**		**Deaths**		**% of Complication**		
≤2		278		30		11%		
>2		84		52		62%		

**Table 8 tropicalmed-07-00007-t008:** Multivariate Cox regression on factors predicting complications *.

Factors			
	Hazard Ratio	95% CI	*p* Value
*RRT requirement			
None/*Peritoneal dialysis	Reference		
Hemodialysis	3.21	1.56 to 6.63	0.0016
Diabetes			
Absent	Reference		
Present	3.21	2.01 to 5.12	0.0001
Dyspnea			
Absent	Reference		
Present	5.13	2.39 to 11.03	0.0001
Hemoptysis			
Absent	Reference		
Present	2.72	1.40 to 5.29	0.0032
Platelet			
Normal	Reference		
Abnormal	2.23	1.20 to 4.16	0.0114
*Urine Output			
Normal/*Oliguric	Reference		
Anuric	1.65	0.94 to 2.91	0.081

*RRT: renal replacement therapy; * Jaundice and Chest X-ray were removed since it becomes nonsignificant during the multivariate run; *Peritoneal dialysis was merged with None (mode of RRT), as it is also nonsignificant; *Oliguric was merged with Normal (Urine), as it is also non-significant; *Urine output was retained since it is still significant at 10% level of significance

## Data Availability

Not Applicable.

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
