# Peer review of "A Scoring Tool to Predict Pulmonary Complications in Severe Leptospirosis with Kidney Failure"

_tropicalmed, 2022, doi:10.3390/tropicalmed7010007_

Round 1

Reviewer 1 Report

I just have one question. 

  • How does the author diagnose the leptospirosis? Leptospirosis is easy to confused with the common cold and flu. Clinical manifestations are not sufficient to diagnose accurately. The laboratory detection is need, such as PCR or MAT. Did the author have laboratory tests for these 380 people?

Author Response

Point 1: How does the author diagnose the leptospirosis? Leptospirosis is easy to confused with the common cold and flu. Clinical manifestations are not sufficient to diagnose accurately. The laboratory detection is need, such as PCR or MAT. Did the author have laboratory tests for these 380 people?

Response 1: Leptospirosis was diagnosed based on clinical manifestations according to the 2010 Philippine Clinical Practice Guidelines on the Diagnosis, Management and Prevention of Leptospirosis. Patients with at least 2 days of acute febrile illness AND either reside in a flooded area with a high risk exposure (wading in flood waters, bitten by rats, work in sewerage areas) AND presenting with at least 2 of the following symptoms: myalgia, calf tenderness, conjunctival suffusion, chills, abdominal pain, headache, jaundice, or oliguria are diagnosed to be suspected with leptospirosis.

Serum leptospiral antibody tests were also done for these patients. Samples were also sent to the Research Institute of Tropical Medicine (RITM) for detection of Leptospira spp DNA via PCR tests. Out of the 380 patients who were clinically diagnosed in our study, 142 were positive IgM leptospiral antibody test, 64 were positive for the microscopic agglutination test and 56 were positive for the real-time PCR detection of pathogenic Leptospira spp.

Reviewer 2 Report

Study from Rizza Antoinette Y. So and colleagues proposed an scoring tool to predict pulmonary complications in severe leptospirosis with kidney failure. This study reviewed 380 patients with severe leptospirosis, and obtained a scoring tool predicting pulmonary complications by logistic regression. This simple clinical score may contribute to rapid formula for clinician that may save more lives. I have little questions for improvement.

  1. line 103-105, the diagnosis of leptospirosis was based on clinical manifestations according to the 2010 Philippine Clinical Practice Guidelines on the Diagnosis, Management and Prevention of Leptospirosis. Whether other diagnostic methods were applied, such as molecular diagnosis, cause the early clinical signs of leptospirosis are similar with some diseases.
  2. Are there any co-infections in 380 leptospirosis patients?
  3. Is there any criteria to select patients above 18 years of age? Why?
  4. line 279-283, the authors listed two published articles with opposite conclusions. Combined with authors’ study, I’d like to read more about this discrepancy.
  5. line 36, “L. Interrogans” should be “L. interrogans”.

Author Response

Point 1: line 103-105, the diagnosis of leptospirosis was based on clinical manifestations according to the 2010 Philippine Clinical Practice Guidelines on the Diagnosis, Management and Prevention of Leptospirosis. Whether other diagnostic methods were applied, such as molecular diagnosis, cause the early clinical signs of leptospirosis are similar with some diseases.

Response 1: Serum leptospiral antibody tests were also done for these patients. Samples were also sent to the Research Institute of Tropical Medicine (RITM) for detection of Leptospira spp DNA via PCR tests.  Out of the 380 patients who were clinically diagnosed in our study, 142 were positive IgM leptospiral antibody test, 64 were positive for the microscopic agglutination test and 56 were positive for the real-time PCR detection of pathogenic Leptospira spp.

Point 2: Are there any co-infections in 380 leptospirosis patients?

Response 2: Based on the review of the researchers, no significant co-infections were detected in the sample population. Out of the 380 leptospirosis patients, 34 had co-existing infections - 18 had UTI, 15 had pneumonia and one had catheter-related bloodstream infection, all with positive urine, sputum and blood cultures, respectively.

Point 3: Is there any criteria to select patients above 18 years of age? Why?

Response 3: The researchers only selected patients above 18 years of age because the study only encompassed the adult population. We did not include the pediatric population in our study.

Point 4: line 279-283, the authors listed two published articles with opposite conclusions. Combined with authors’ study, I’d like to read more about this discrepancy.

Response 4:

This study reports a 14% mortality rate in the 380 leptospirosis patients included in the study. Among those with pulmonary complications, 56.5% died. Upon review of related literature, a Brazil study by Marotto et al in 2010 reported an overall case fatality rate of 8% in 203 patients who developed Leptospirosis-associated pulmonary hemorrhage syndrome (LPHS) from 1998 to 2004.[1] In a 2011 study by Paganin et al between 1996 and 2006 in the Reunion Islands, overall mortality was 15.7% (21 patients) out of 134 leptospirosis patients with pulmonary complications.  Forty patients had severe pulmonary leptospirosis with ARF and 16 of them died (40%). [2]  Our Philippine study had a higher mortality rate as compared to these two countries. This is attributed to our higher number of leptospirosis patients during outbreaks during the rainy season. Our single center itself was able to record 380 leptospirosis patients in merely a year.

Since the NKTI is also a tertiary referral center, most leptospirosis patients admitted to our institution already have severe leptospirosis, most presenting with both pulmonary complications and dialysis-requiring acute kidney injury (AKI). In the Brazil study, there was note of significantly increased serum creatinine in the patients with LPHS and this was identified as an independent risk factor for developing LPHS but there was no mention if their patients presented with oliguria, anuria, or underwent dialysis.[1] In the Reunion Islands study, they were able to shed more light about AKI in leptospirosis patients with pulmonary complications. Their study showed that 44 (33%) of the leptospirosis patients with pulmonary involvement had oliguria/anuria and this was found to be an independent factor related to severe pulmonary involvement. They postulated that the combination of AKI with respiratory impairment presented a risk of lethal evolution: mortality is 18% for AKI and 24% for respiratory impairment alone, but 55% when the two are combined.[2]

  1. Marotto PC, Ko AI, Murta-Nascimento C, et al. Early identification of leptospirosis-associated pulmonary hemorrhage syndrome by use of a validated prediction model. J Infect. 2010;60(3):218-223.
  2. Paganin F, Bourdin A, Borgherini G, et al. Manifestations pulmonaires de la leptospirose [Pulmonary manifestations of leptospirosis]. Rev Mal Respir. 2009;26(9):971-979.

Point 5: line 36, “L. Interrogans” should be “L. interrogans”.

Response 5: Noted on this correction

Reviewer 3 Report

Dear Authors,

The current manuscript presents an important retrospective study. The results are well-explained.

I have a few concerns:

  1. Why is the study heavily male-biased? The authors have mentioned that there were more males than females in the study but what is the reason behind it? Are men at higher risk to be affected by flooding?
  2. It is rather surprising that (as shown in Table 1), pulmonary complications were higher in subjects between 18 and 40 years (54%) vs 41-60 (36.5%) and 61 and above years (9.4%) given that diabetes is a major risk factor. Among the diabetics with pulmonary complications, what was the age distribution?
  3. It is surprising that abnormality in WBC/leukocytes' counts are not significant with pulmonary complications caused by a bacterial disease. How do the authors explain this?
  4. Potassium and phosphorous seem to be statistically significant as a factor for pulmonary complication. How do the authors explain this?

Author Response

Point 1: Why is the study heavily male-biased? The authors have mentioned that there were more males than females in the study but what is the reason behind it? Are men at higher risk to be affected by flooding?

Response 1: Majority of the patients included in the study were male because there is an increased chance for males to be exposed to the etiologic agent for leptospirosis due to occupational risk. Most of our patients have high-risk occupations such as drivers, market/sidewalk vendors, garbage collectors, security guards, messengers, farmers, carpenters, plumbers, construction workers, and fishermen which are usually male-dominated jobs.

Point 2: It is rather surprising that (as shown in Table 1), pulmonary complications were higher in subjects between 18 and 40 years (54%) vs 41-60 (36.5%) and 61 and above years (9.4%) given that diabetes is a major risk factor. Among the diabetics with pulmonary complications, what was the age distribution?

Response 2: We attribute the fact that pulmonary complications are higher in the 18-40 year old age group because 60% of the population in this study belong to this age bracket, as compared to the 41-60 age group (34%) and the 61 and above age group (6%). Among the diabetic with pulmonary complications (40%), 40% belong to the 18-40 year old age group, 47% are between 41-60 years of age and only 13% are aged 61 years old and above. Again, these are attributed to the fact that the subjects in this study group are mostly young and able bodied young males between the ages of 20-40 years old who belong to the labor force and work in rat infested areas.

Point 3: It is surprising that abnormality in WBC/leukocytes' counts are not significant with pulmonary complications caused by a bacterial disease. How do the authors explain this?

Response 3: Most patients with pulmonary complications have abnormal WBC counts but results were not statistically significant. According to studies, in the majority of patients, the white blood cell (WBC) count will be within normal limits. Leukocytosis occurs in a significant minority, approximately 35-40% of patients.  In the studies by Marotto and Paganin et al, their data showed that WBC count was not significant risk factor to develop pulmonary complications. There was no postulated reason for this in their papers.[1,2]

  1. Marotto PC, Ko AI, Murta-Nascimento C, et al. Early identification of leptospirosis-associated pulmonary hemorrhage syndrome by use of a validated prediction model. J Infect. 2010;60(3):218-223.
  2. Paganin F, Bourdin A, Borgherini G, et al. Manifestations pulmonaires de la leptospirose [Pulmonary manifestations of leptospirosis]. Rev Mal Respir. 2009;26(9):971-979.

Point 4: Potassium and phosphorous seem to be statistically significant as a factor for pulmonary complication. How do the authors explain this?

Response 4: Potassium and phosphorus abnormalities are attributed to renal dysfunction and are indicators of tubular damage seen in AKI. Related studies have shown that oliguria/anuria, which are signs of AKI, are independent risk factors to developing pulmonary complications in leptospirosis.

Round 2

Reviewer 1 Report

Accept

Author Response

Thank you very much for all your suggestions.